# Multi-Task Zipping via Layer-wise Neuron Sharing

**Xiaoxi He**
ETH Zurich
hex@ethz.ch

**Zimu Zhou**∗
ETH Zurich
zzhou@tik.ee.ethz.ch

**Lothar Thiele**
ETH Zurich
thiele@ethz.ch

## Abstract

Future mobile devices are anticipated to perceive, understand and react to the world on their own by running multiple correlated deep neural networks on-device. Yet the complexity of these neural networks needs to be trimmed down both within-model and cross-model to fit in mobile storage and memory. Previous studies squeeze the redundancy within a single model. In this work, we aim to reduce the redundancy across multiple models. We propose Multi-Task Zipping (MTZ), a framework to automatically merge correlated, pre-trained deep neural networks for cross-model compression. Central in MTZ is a layer-wise neuron sharing and incoming weight updating scheme that induces a minimal change in the error function. MTZ inherits information from each model and demands light retraining to re-boost the accuracy of individual tasks. Evaluations show that MTZ is able to fully merge the hidden layers of two VGG-16 networks with a 3.18% increase in the test error averaged on ImageNet and CelebA, or share 39.61% parameters between the two networks with $< 0.5\%$ increase in the test errors for both tasks. The number of iterations to retrain the combined network is at least $17.8\times$ lower than that of training a single VGG-16 network. Moreover, experiments show that MTZ is also able to effectively merge multiple residual networks.

## 1 Introduction

AI-powered mobile applications increasingly demand *multiple* deep neural networks for *correlated* tasks to be performed continuously and concurrently on resource-constrained devices such as wearables, smartphones, self-driving cars, and drones [5, 18]. While many pre-trained models for different tasks are available [14, 23, 25], it is often infeasible to deploy them directly on mobile devices. For instance, VGG-16 models for object detection [25] and facial attribute classification [17] both contain over 130M parameters. Packing multiple such models easily strains mobile storage and memory. Sharing information among tasks holds potential to reduce the sizes of multiple correlated models without incurring drop in individual task inference accuracy.

We study information sharing in the context of *cross-model compression*, which seeks *effective* and *efficient* information sharing mechanisms among *pre-trained* models for multiple tasks to reduce the size of the combined model without accuracy loss in each task. A solution to cross-model compression is multi-task learning (MTL), a paradigm that jointly learns multiple tasks to improve the robustness and generalization of tasks [1, 5]. However, most MTL studies use heuristically configured shared structures, which may lead to dramatic accuracy loss due to improper sharing of knowledge [31]. Some recent proposals [17, 19, 28] automatically decide "what to share" in deep neural networks. Yet deep MTL usually involves enormous training overhead [31]. Hence it is inefficient to ignore the already trained parameters in each model and apply MTL for cross-model compression.

We propose Multi-Task Zipping (MTZ), a framework to automatically and adaptively merge correlated, well-trained deep neural networks for cross-model compression via neuron sharing. It decides the

---

∗Corresponding Author: Zimu Zhou.

optimal sharable pairs of neurons on a layer basis and adjusts their incoming weights such that minimal errors are introduced in each task. Unlike MTL, MTZ inherits the parameters of each model and optimizes the information to be shared among models such that only light retraining is necessary to resume the accuracy of individual tasks. In effect, it squeezes the *inter-network redundancy* from multiple already trained deep neural networks. With appropriate hardware support, MTZ can be further integrated with existing proposals for *single-model compression*, which reduce the *intra-network redundancy* via pruning [4, 6, 8, 15] or quantization [2, 7].

The contributions and results of this work are as follows.

- We propose MTZ, a framework that automatically merges multiple correlated, pre-trained deep neural networks. It squeezes the task relatedness across models via layer-wise neuron sharing, while requiring light retraining to re-boost the accuracy of the combined model.

- Experiments show that MTZ is able to merge all the hidden layers of two LeNet networks [14] (differently trained on MNIST) without increase in test errors. MTZ manages to share 39.61% parameters between the two VGG-16 networks pre-trained for object detection (on ImageNet [24]) and facial attribute classification (on CelebA [16]), while incurring less than 0.5% increase in test errors. Even when all the hidden layers are fully merged, there is a moderate (averaged 3.18%) increase in test errors for both tasks. MTZ achieves the above performance with at least $17.9\times$ fewer iterations than training a single VGG-16 network from scratch [25]. In addition, MTZ is able to share 90% of the parameters among five ResNets on five different visual recognition tasks while inducing negligible loss on accuracy.

## 2   Related Work

**Multi-task Learning.** Multi-task learning (MTL) leverages the task relatedness in the form of shared structures to jointly learn multiple tasks [1]. Our MTZ resembles MTL in effect, *i.e.*, sharing structures among related tasks, but differs in objectives. MTL jointly trains multiple tasks to improve their generalization, while MTZ aims to compress multiple *already trained* tasks with mild training overhead. Georgiev *et al.* [5] are the first to apply MTL in the context of multi-model compression. However, as in most MTL studies, the shared topology is heuristically configured, which may lead to improper knowledge transfer [29]. Only a few schemes optimize *what to share among tasks*, especially for deep neural networks. Yang *et al.* propose to learn cross-task sharing structure at each layer by tensor factorization [28]. Cross-stitching networks [19] learn an optimal shared and task-specific representations using cross-stitch units. Lu *et al.* automatically grow a wide multi-task network architecture from a thin network by branching [17]. Similarly, Rebuffi *et al.* sequentially add new tasks to a main task using residual adapters for ResNets [21]. Different to the above methods, MTZ inherits the parameters directly from each pre-trained network when optimizing the neurons shared among tasks in each layer and demands light retraining.

**Single-Model Compression.** Deep neural networks are typically over-parameterized [3]. There have been various model compression proposals to reduce the redundancy in a *single* neural network. Pruning-based methods sparsify a neural network by eliminating unimportant weights (connections) [4, 6, 8, 15]. Other approaches reduce the dimensions of a neural network by neuron trimming [11] or learning a compact (yet dense) network via knowledge distillation [22, 10]. The memory footprint of a neural network can be further reduced by lowering the precision of parameters [2, 7].Unlike previous research that deals with the *intra-redundancy* of a single network, our work reduces the *inter-redundancy* among multiple networks. In principle, our method is a dimension reduction based *cross-model* compression scheme via neuron sharing. Although previous attempts designed for a single network may apply, they either adopt heuristic neuron similarity criterion [11] or require training a new network from scratch [22, 10]. Our neuron similarity metric is grounded upon parameter sensitivity analysis for neural networks, which is applied in single-model weight pruning [4, 8, 15]. Our work can be integrated with single-model compression to further reduce the size of the combined network.

## 3   Layer-wise Network Zipping

### 3.1   Problem Statement

Consider two inference tasks $A$ and $B$ with the corresponding two *well-trained* models $M^A$ and $M^B$, *i.e.*, trained to a local minimum in error. Our goal is to construct a combined model $M^C$ by sharing

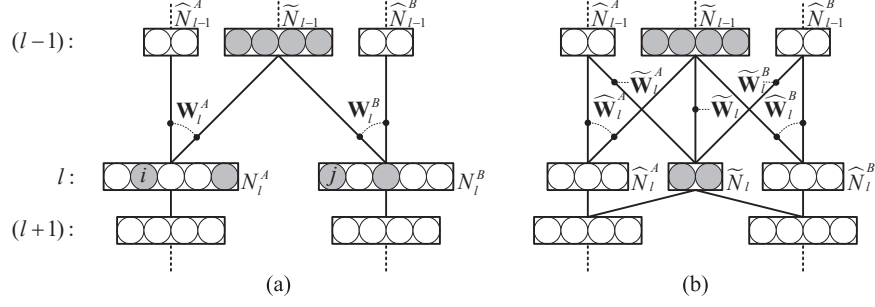

Figure 1: An illustration of layer zipping via neuron sharing: neurons and the corresponding weight matrices (a) before and (b) after zipping the $l$-th layers of $M^A$ and $M^B$.

as many neurons between layers in $M^A$ and $M^B$ as possible such that *(i)* $M^C$ has minimal loss in inference accuracy for the two tasks and *(ii)* the construction of $M^C$ involves minimal retraining.

For ease of presentation, we explain our method with two feed-forward networks of dense fully connected (FC) layers. We extend MTZ to convolutional (CONV) layers in Sec. 3.5, sparse layers in Sec. 3.6 and residual networks (ResNets) in Sec. 3.7. We assume the same input domain and the same number of layers in $M^A$ and $M^B$.

## 3.2 Layer Zipping via Neuron Sharing: Fully Connected Layers

This subsection presents the procedure of zipping the $l$-th layers ($1 \leq l \leq L-1$) in $M^A$ and $M^B$ given the previous $(l-1)$ layers have been merged (see Fig. 1). We denote the input layers as the 0-th layers. The $L$-th layers are the output layers of $M^A$ and $M^B$.

Denote the weight matrices of the $l$-th layers in $M^A$ and $M^B$ as $\mathbf{W}_l^A \in \mathbb{R}^{N_{l-1}^A \times N_l^A}$ and $\mathbf{W}_l^B \in \mathbb{R}^{N_{l-1}^B \times N_l^B}$, where $N_l^A$ and $N_l^B$ are the numbers of neurons in the $l$-th layers in $M^A$ and $M^B$. Assume $\tilde{N}_{l-1} \in [0, \min\{N_{l-1}^A, N_{l-1}^B\}]$ neurons are shared between the $(l-1)$-th layers in $M^A$ and $M^B$. Hence there are $\hat{N}_{l-1}^A = N_{l-1}^A - \tilde{N}_{l-1}$ and $\hat{N}_{l-1}^B = N_{l-1}^B - \tilde{N}_{l-1}$ task-specific neurons left in the $(l-1)$-th layers in $M^A$ and $M^B$, respectively.

**Neuron Sharing.** To enforce neuron sharing between the $l$-th layers in $M^A$ and $M^B$, we calculate the *functional difference* between the $i$-th neuron in layer $l$ in $M^A$, and the $j$-th neuron in the same layer in $M^B$. The functional difference is measured by a metric $d[\tilde{\mathbf{w}}_{l,i}^A, \tilde{\mathbf{w}}_{l,j}^B]$, where $\tilde{\mathbf{w}}_{l,i}^A, \tilde{\mathbf{w}}_{l,j}^B \in \mathbb{R}^{\tilde{N}_{l-1}}$ are the incoming weights of the two neurons from the *shared* neurons in the $(l-1)$-th layer. We do not alter incoming weights from the non-shared neurons in the $(l-1)$-th layer because they are likely to contain task-specific information only.

To zip the $l$-th layers in $M^A$ and $M^B$, we first calculate the functional difference for each pair of neurons $(i, j)$ in layer $l$ and select $\tilde{N}_l \in [0, \min\{N_l^A, N_l^B\}]$ pairs with the smallest functional difference. These pairs of neurons form a set $\{(i_k, j_k)\}$, where $k = 0, \cdots, \tilde{N}_l$ and each pair is merged into one neuron. Thus the neurons in the $l$-th layers in $M^A$ and $M^B$ fall into three groups: $\tilde{N}_l$ shared, $\hat{N}_l^A = N_l^A - \tilde{N}_l$ specific for $A$ and $\hat{N}_l^B = N_l^B - \tilde{N}_l$ specific for $B$.

**Weight Matrices Updating.** Finally the weight matrices $\mathbf{W}_l^A$ and $\mathbf{W}_l^B$ are re-organized as follows. The weights vectors $\tilde{\mathbf{w}}_{l,i_k}^A$ and $\tilde{\mathbf{w}}_{l,j_k}^B$, where $k = 0, \cdots, \tilde{N}_l$, are merged and replaced by a matrix $\tilde{\mathbf{W}}_l \in \mathbb{R}^{\tilde{N}_{l-1} \times \tilde{N}_l}$, whose columns are $\tilde{\mathbf{w}}_{l,k} = f(\tilde{\mathbf{w}}_{l,i_k}^A, \tilde{\mathbf{w}}_{l,j_k}^B)$, where $f(\cdot)$ is an *incoming weight update function*. $\tilde{\mathbf{W}}_l$ represents the task-relatedness between $A$ and $B$ from layer $(l-1)$ to layer $l$. The incoming weights from the $\hat{N}_{l-1}^A$ neurons in layer $(l-1)$ to the $\hat{N}_l^A$ neurons in layer $l$ in $M^A$ form a matrix $\hat{\mathbf{W}}_l^A \in \mathbb{R}^{N_{l-1}^A \times \hat{N}_l^A}$. The remaining columns in $\mathbf{W}_l^A$ are packed as $\tilde{\mathbf{W}}_l^A \in \mathbb{R}^{\hat{N}_{l-1}^A \times \tilde{N}_l}$. Matrices $\hat{\mathbf{W}}_l^A$ and $\tilde{\mathbf{W}}_l^A$ contain the task-specific information for $A$ between layer $(l-1)$ and layer $l$. For task $B$, we organize matrices $\hat{\mathbf{W}}_l^B \in \mathbb{R}^{N_{l-1}^B \times \hat{N}_l^B}$ and $\tilde{\mathbf{W}}_l^B \in \mathbb{R}^{\hat{N}_{l-1}^B \times \tilde{N}_l}$ in a similar manner. We also adjust the order of rows in the weight matrices in the $(l+1)$-th layers, $\mathbf{W}_{l+1}^A$ and $\mathbf{W}_{l+1}^B$, to maintain the correct connections among neurons.

The above layer zipping process can reduce $\tilde{N}_{l-1} \times \tilde{N}_l$ weights from $\mathbf{W}_l^A$ and $\mathbf{W}_l^B$. Essential in MTZ are the neuron functional difference metric $d[\cdot]$ and the incoming weight update function $f(\cdot)$. They are designed to demand only light retraining to recover the original accuracy.

### 3.3 Neuron Functional Difference and Incoming Weight Update

This subsection introduces our neuron functional difference metric $d[\cdot]$ and weight update function $f(\cdot)$ leveraging previous research on parameter sensitivity analysis for neural networks [4, 8, 15].

**Preliminaries.** A naive approach to accessing the impact of a change in some parameter vector $\boldsymbol{\theta}$ on the objective function (training error) $E$ is to apply the parameter change and re-evaluate the error on the entire training data. An alternative is to exploit second order derivatives [4, 8]. Specifically, the Taylor series of the change $\delta E$ in training error due to certain parameter vector change $\delta\boldsymbol{\theta}$ is [8]:

$$\delta E = \left(\frac{\partial E}{\partial \boldsymbol{\theta}}\right)^{\top} \cdot \delta\boldsymbol{\theta} + \frac{1}{2}\delta\boldsymbol{\theta}^{\top} \cdot \mathbf{H} \cdot \delta\boldsymbol{\theta} + O(\|\delta\boldsymbol{\theta}\|^3) \tag{1}$$

where $\mathbf{H} = \partial^2 E/\partial\boldsymbol{\theta}^2$ is the Hessian matrix containing all the second order derivatives. For a network trained to a local minimum in $E$, the first term vanishes. The third and higher order terms can also be ignored [8]. Hence:

$$\delta E = \frac{1}{2}\delta\boldsymbol{\theta}^{\top} \cdot \mathbf{H} \cdot \delta\boldsymbol{\theta} \tag{2}$$

Eq.(2) approximates the deviation in error due to parameter changes. However, it is still a bottleneck to compute and store the Hessian matrix $\mathbf{H}$ of a modern deep neural network. As next, we harness the trick in [4] to break the calculations of Hessian matrices into layer-wise, and propose a Hessian-based neuron difference metric as well as the corresponding weight update function for neuron sharing.

**Method.** Inspired by [4] we define the error functions of $M^A$ and $M^B$ in layer $l$ as

$$E_l^A = \frac{1}{n_A}\sum \|\tilde{\mathbf{y}}_l^A - \mathbf{y}_l^A\|^2 \tag{3}$$

$$E_l^B = \frac{1}{n_B}\sum \|\tilde{\mathbf{y}}_l^B - \mathbf{y}_l^B\|^2 \tag{4}$$

where $\mathbf{y}_l^A$ and $\tilde{\mathbf{y}}_l^A$ are the *pre-activation* outputs of the $l$-th layers in $M^A$ before and after layer zipping, evaluated on one instance from the training set of $A$; $\mathbf{y}_l^B$ and $\tilde{\mathbf{y}}_l^B$ are defined in a similar way; $\|\cdot\|$ is $l^2$-norm; $n_A$ and $n_B$ are the number of training samples for $M^A$ and $M^B$, respectively; $\Sigma$ is the summation over all training instances. Since $M^A$ and $M^B$ are trained to a local minimum in training error, $E_l^A$ and $E_l^B$ will have the same minimum points as the corresponding training errors.

We further define an error function of the combined network in layer $l$ as

$$E_l = \alpha E_l^A + (1-\alpha)E_l^B \tag{5}$$

where $\alpha \in (0,1)$ is used to balance the errors of $M^A$ and $M^B$. The change in $E_l$ with respect to neuron sharing in the $l$-th layer can be expressed in a similar form as Eq.(2):

$$\delta E_l = \frac{1}{2}(\delta\tilde{\mathbf{w}}_{l,i}^A)^{\top} \cdot \tilde{\mathbf{H}}_{l,i}^A \cdot \delta\tilde{\mathbf{w}}_{l,i}^A + \frac{1}{2}(\delta\tilde{\mathbf{w}}_{l,j}^B)^{\top} \cdot \tilde{\mathbf{H}}_{l,j}^B \cdot \delta\tilde{\mathbf{w}}_{l,j}^B \tag{6}$$

where $\delta\tilde{\mathbf{w}}_{l,i}^A$ and $\delta\tilde{\mathbf{w}}_{l,j}^B$ are the adjustments in the weights of $i$ and $j$ to merge the two neurons; $\tilde{\mathbf{H}}_{l,i}^A = \partial^2 E_l/(\partial\tilde{\mathbf{w}}_{l,i}^A)^2$ and $\tilde{\mathbf{H}}_{l,j}^B = \partial^2 E_l/(\partial\tilde{\mathbf{w}}_{l,j}^B)^2$ denote the *layer-wise* Hessian matrices. Similarly to [4], the layer-wise Hessian matrices can be calculated as

$$\tilde{\mathbf{H}}_{l,i}^A = \frac{\alpha}{n_A}\sum \mathbf{x}_{i-1}^A \cdot (\mathbf{x}_{i-1}^A)^{\top} \tag{7}$$

$$\tilde{\mathbf{H}}_{l,j}^B = \frac{1-\alpha}{n_B}\sum \mathbf{x}_{j-1}^B \cdot (\mathbf{x}_{j-1}^B)^{\top} \tag{8}$$

where $\mathbf{x}_{i-1}^A$ and $\mathbf{x}_{j-1}^B$ are the outputs of the merged neurons from layer $(l-1)$ in $M^A$ and $M^B$, respectively.

When sharing the $i$-th and $j$-th neurons in the $l$-th layers of $M^A$ and $M^B$, respectively, our aim is to minimize $\delta E_l$, which can be formulated as the optimization problem below:

$$\min_{(i,j)} \{ \min_{(\delta\tilde{\mathbf{w}}^A_{l,i}, \delta\tilde{\mathbf{w}}^B_{l,j})} \delta E_l \} \text{ s.t. } \tilde{\mathbf{w}}^A_{l,i} + \delta\tilde{\mathbf{w}}^A_{l,i} = \tilde{\mathbf{w}}^B_{l,j} + \delta\tilde{\mathbf{w}}^B_{l,j} \tag{9}$$

Applying the method of Lagrange multipliers, the optimal weight changes and the resulting $\delta E_l$ are:

$$\delta\tilde{\mathbf{w}}^{A,opt}_{l,i} = (\tilde{\mathbf{H}}^A_{l,i})^{-1} \cdot \left( (\tilde{\mathbf{H}}^A_{l,i})^{-1} + (\tilde{\mathbf{H}}^B_{l,j})^{-1} \right)^{-1} \cdot (\tilde{\mathbf{w}}^B_{l,j} - \tilde{\mathbf{w}}^A_{l,i}) \tag{10}$$

$$\delta\tilde{\mathbf{w}}^{B,opt}_{l,j} = (\tilde{\mathbf{H}}^B_{l,j})^{-1} \cdot \left( (\tilde{\mathbf{H}}^A_{l,i})^{-1} + (\tilde{\mathbf{H}}^B_{l,j})^{-1} \right)^{-1} \cdot (\tilde{\mathbf{w}}^A_{l,i} - \tilde{\mathbf{w}}^B_{l,j}) \tag{11}$$

$$\delta E^{opt}_l = \frac{1}{2} (\tilde{\mathbf{w}}^A_{l,i} - \tilde{\mathbf{w}}^B_{l,j})^\top \cdot \left( (\tilde{\mathbf{H}}^A_{l,i})^{-1} + (\tilde{\mathbf{H}}^B_{l,j})^{-1} \right)^{-1} \cdot (\tilde{\mathbf{w}}^A_{l,i} - \tilde{\mathbf{w}}^B_{l,j}) \tag{12}$$

Finally, we define the neuron functional difference metric $d[\tilde{\mathbf{w}}^A_{l,i}, \tilde{\mathbf{w}}^B_{l,j}] = \delta E^{opt}_l$, and the weight update function $f(\tilde{\mathbf{w}}^A_{l,i}, \tilde{\mathbf{w}}^B_{l,j}) = \tilde{\mathbf{w}}^A_{l,i} + \delta\tilde{\mathbf{w}}^{A,opt}_{l,i} = \tilde{\mathbf{w}}^B_{l,j} + \delta\tilde{\mathbf{w}}^{B,opt}_{l,j}$.

### 3.4 MTZ Framework

Algorithm 1 outlines the process of MTZ on two tasks of the same input domain, *e.g.*, images. We first construct a joint input layer. In case the input layer dimensions are not equal in both tasks, the dimension of the joint input layer equals the larger dimension of the two original input layers, and fictive connections (*i.e.*, weight 0) are added to the model whose original input layers are smaller. Afterwards we begin layer-wise neuron sharing and weight matrix updating from the first hidden layer. The two networks are "zipped" layer by layer till the last hidden layer and we obtain a combined network. After merging each layer, the networks are retrained to re-boost the accuracy.

**Practical Issues.** We make the following notes on the practicability of MTZ.

- *How to set the number of neurons to be shared?* One can directly set $\tilde{N}_l$ neurons to be shared for the $l$-th layers, or set a layer-wise threshold $\varepsilon_l$ instead. Given a threshold $\varepsilon_l$, MTZ shares pairs of neurons where $\{(i_k, j_k) | d[\tilde{\mathbf{w}}^A_{l,i_k}, \tilde{\mathbf{w}}^B_{l,j_k}] < \varepsilon_l\}$. In this case $\tilde{N}_l = |\{(i_k, j_k)\}|$. One can set $\{\tilde{N}_l\}$ if there is a hard constraint on storage or memory. Otherwise $\{\varepsilon_l\}$ can be set if accuracy is of higher priority. Note that $\{\varepsilon_l\}$ controls the layer-wise error $\delta E_l$, which correlates to the accumulated errors of the outputs in layer $L$ $\tilde{\varepsilon}^A = \frac{1}{\sqrt{n_A}} \sum \|\tilde{\mathbf{x}}^A_L - \mathbf{x}^A_L\|$ and $\tilde{\varepsilon}^B = \frac{1}{\sqrt{n_B}} \sum \|\tilde{\mathbf{x}}^B_L - \mathbf{x}^B_L\|$ [4].

- *How to execute the combined model for each task?* During inference, only task-related connections in the combined model are enabled. For instance, when performing inference on task $A$, we only activate $\{\hat{\mathbf{W}}^A_l\}$, $\{\tilde{\mathbf{W}}^A_l\}$ and $\{\tilde{\mathbf{W}}_l\}$, while $\{\tilde{\mathbf{W}}^B_l\}$ and $\{\hat{\mathbf{W}}^B_l\}$ are disabled (*e.g.*, by setting them to zero).

- *How to zip more than two neural networks?* MTZ is able to zip more than two models by sequentially adding each network into the joint network, and the calculated Hessian matrices of the already zipped joint network can be reused. Therefore, MTZ is scalable in regards to both the depth of each network and the number of tasks to be zipped. Also note that since calculating the Hessian matrix of one layer requires only its layer input, only one forward pass in total from each model is needed for the merging process (excluding retraining).

### 3.5 Extension to Convolutional Layers

The layer zipping procedure of two convolutional layers are very similar to that of two fully connected layers. The only difference is that sharing is performed on kernels rather than neurons. Take the $i$-th kernel of size $k_l \times k_l$ in layer $l$ of $M^A$ as an example. Its incoming weights from the previous shared kernels are $\tilde{\mathbf{W}}^{A,in}_{l,i} \in \mathbb{R}^{k_l \times k_l \times \tilde{N}_{l-1}}$. The weights are then flatten into a vector $\tilde{\mathbf{w}}^A_{l,i}$ to calculate functional differences. As with in Sec. 3.2, after layer zipping in the $l$-th layers, the weight matrices in the $(l+1)$-th layers need careful permutations regarding the flattening ordering to maintain correct connections among neurons, especially when the next layers are fully connected layers.

**Algorithm 1:** Multi-task Zipping via Layer-wise Neuron Sharing

**input** : $\{\mathbf{W}_l^A\}, \{\mathbf{W}_l^B\}$: weight matrices of $M^A$ and $M^B$
        $\mathbf{X}^A, \mathbf{X}^B$: training datum of task A and B (including labels)
        $\alpha$: coefficient to adjust the zipping balance of $M^A$ and $M^B$
        $\{\tilde{N}_l\}$: number of neurons to be shared in layer $l$

1 **for** $l = 1, \ldots, L-1$ **do**
2     Calculate inputs for the current layer $\mathbf{x}_{l-1}^A$ and $\mathbf{x}_{l-1}^B$ using training data from $\mathbf{X}^A$ and $\mathbf{X}^B$
     and forward propagation
3     $\tilde{\mathbf{H}}_{l,i}^A \leftarrow \frac{\alpha}{n_A} \sum \mathbf{x}_{i-1}^A \cdot (\mathbf{x}_{i-1}^A)^\top$
4     $\tilde{\mathbf{H}}_{l,j}^B \leftarrow \frac{1-\alpha}{n_B} \sum \mathbf{x}_{j-1}^B \cdot (\mathbf{x}_{j-1}^B)^\top$
5     Select $\tilde{N}_l$ pairs of neurons $\{(i_k, j_k)\}$ with the smallest $d[\tilde{\mathbf{w}}_{l,i}^A, \tilde{\mathbf{w}}_{l,j}^B]$
6     **for** $k \leftarrow 1, \ldots, \tilde{N}_l$ **do**
7         $\tilde{\mathbf{w}}_{l,k} \leftarrow f(\tilde{\mathbf{w}}_{l,i_k}^A, \tilde{\mathbf{w}}_{l,j_k}^B)$
8     Re-organize $\mathbf{W}_l^A$ and $\mathbf{W}_l^B$ into $\tilde{\mathbf{W}}_l, \hat{\mathbf{W}}_l^A, \tilde{\mathbf{W}}_l^A, \hat{\mathbf{W}}_l^B$ and $\tilde{\mathbf{W}}_l^B$
9     Permute the order of rows in $\mathbf{W}_{l+1}^A$ and $\mathbf{W}_{l+1}^B$ to maintain correct connections
10     Conduct a light retraining on task $A$ and $B$ to re-boost accuracy of the joint model

**output** : $\{\hat{\mathbf{W}}_l^A\}, \{\tilde{\mathbf{W}}_l^A\}, \{\tilde{\mathbf{W}}_l\}, \{\tilde{\mathbf{W}}_l^B\}, \{\hat{\mathbf{W}}_l^B\}$: weights of the zipped multi-task model $M^C$

### 3.6 Extension to Sparse Layers

Since the pre-trained neural networks may have already been sparsified via weight pruning, we also extend MTZ to support sparse models. Specifically, we use sparse matrices, where zeros indicate no connections, to represent such sparse models. Then the incoming weights from the previous shared neurons/kernels $\tilde{\mathbf{w}}_{l,i}^A, \tilde{\mathbf{w}}_{l,j}^B$ still have the same dimension. Therefore $d[\tilde{\mathbf{w}}_{l,i}^A, \tilde{\mathbf{w}}_{l,j}^B], f(\tilde{\mathbf{w}}_{l,i}^A, \tilde{\mathbf{w}}_{l,j}^B)$ can be calculated as usual. However, we also calculate two mask vectors $\tilde{\mathbf{m}}_{l,i}^A$ and $\tilde{\mathbf{m}}_{l,j}^B$, whose elements are 0 when the corresponding elements in $\tilde{\mathbf{w}}_{l,i}^A$ and $\tilde{\mathbf{w}}_{l,j}^B$ are 0, and 1 otherwise. We pick the mask vector with more $1's$ and apply it to $\tilde{\mathbf{w}}_l$. This way the combined model always have a smaller number of connections (weights) than the sum of the original two models.

### 3.7 Extension to Residual Networks

MTZ can also be extended to merge residual networks [9]. To simplify the merging process, we assume that the last layer is always fully-merged when merging the next layers. Hence after merging we have only matrices $\hat{\mathbf{W}}_l^A \in \mathbb{R}^{N_{l-1}^A \times \hat{N}_l^A}$, $\hat{\mathbf{W}}_l^B \in \mathbb{R}^{N_{l-1}^B \times \hat{N}_l^B}$, and $\tilde{\mathbf{W}}_l \in \mathbb{R}^{\min\{N_{l-1}^A, N_{l-1}^B\} \times \tilde{N}_l}$. This assumption is able to provide decent performance (see Sec. 4.3). Note that the sequence of the channels of the shortcuts need to be permuted before and after the adding operation at the end of each residual block in order to maintain correct connections after zipping.

## 4 Experiments

We evaluate the performance of MTZ on zipping networks pre-trained for the same task (Sec. 4.1) and different tasks (Sec. 4.2 and Sec. 4.3). We mainly assess the test errors of each task after network zipping and the retraining overhead involved. MTZ is implemented with TensorFlow.All experiments are conducted on a workstation equipped with Nvidia Titan X (Maxwell) GPU.

### 4.1 Performance to Zip Two Networks (LeNet) Pre-trained for the Same Task

This experiment validates the effectiveness of MTZ by merging two differently trained models for the same task. Ideally, two models trained to different local optimums should function the same on the test data. Therefore their hidden layers can be fully merged without incurring any accuracy loss. This experiment aims to show that, by finding the correct pairs of neurons which shares the same functionality, MTZ can achieve the theoretical limit of compression ratio *i.e.*, 100%, even without any retraining involved.

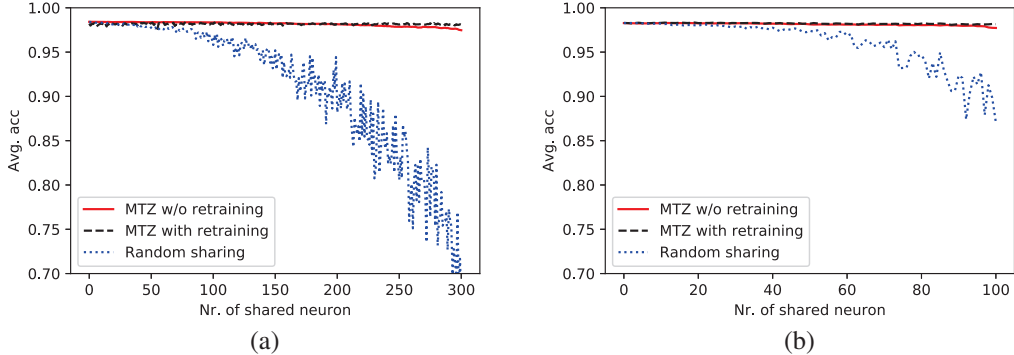

Figure 2: Test error on MNIST by continually sharing neurons in (a) the first and (b) the second fully connected layers of two dense LeNet-300-100 networks till the merged layers are fully shared.

Table 1: Test errors on MNIST by sharing all neurons in two LeNet networks.

| Model | $\text{err}_A$ | $\text{err}_B$ | re-$\text{err}_C$ | # re-iter |
|---|---|---|---|---|
| LeNet-300-100-Dense | 1.57% | 1.60% | 1.64% | 550 |
| LeNet-300-100-Sparse | 1.80% | 1.81% | 1.83% | 800 |
| LeNet-5-Dense | 0.89% | 0.95% | 0.93% | 600 |
| LeNet-5-Sparse | 1.27% | 1.28% | 1.29% | 1200 |

**Dataset and Settings.** We experiment on MNIST dataset with the LeNet-300-100 and LeNet-5 networks [14] to recognize handwritten digits from zero to nine. LeNet-300-100 is a fully connected network with two hidden layers (300 and 100 neurons each), reporting an error from 1.6% to 1.76% on MNIST [4][14]. LeNet-5 is a convolutional network with two convolutional layers and two fully connected layers, which achieves an error ranging from 0.8% to 1.27% on MNIST [4][14].

We train two LeNet-300-100 networks of our own with errors of 1.57% and 1.60%; and two LeNet-5 networks with errors of 0.89% and 0.95%. All the networks are initialized randomly with different seeds, and the training data are also shuffled before every training epoch. After training, the ordering of neurons/kernels in all hidden layers is once more randomly permuted. Therefore the models have completely different parameters (weights). The training of LeNet-300-100 and LeNet-5 networks requires $1.05 \times 10^4$ and $1.1 \times 10^4$ iterations in average, respectively.

For sparse networks, we apply one iteration of L-OBS [4] to prune the weights of the four LeNet networks. We then enforce all neurons to be shared in each hidden layer of the two dense LeNet-300-100 networks, sparse LeNet-300-100 networks, dense LeNet-5 networks, and sparse LeNet-5 networks, using MTZ.

**Results.** Fig. 2a plots the average error after sharing different amounts of neurons in the first layers of two dense LeNet-300-100 networks. Fig. 2b shows the error by further merging the second layers. We compare MTZ with a random sharing scheme, which shares neurons by first picking $(i_k, j_k)$ at random, and then choosing randomly between $\tilde{\mathbf{w}}^A_{l,i_k}$ and $\tilde{\mathbf{w}}^B_{l,j_k}$ as the shared weights $\tilde{\mathbf{w}}_{l_k}$. When all the 300 neurons in the first hidden layers are shared, there is an increase of 0.95% in test error (averaged over the two models) even without retraining, while random sharing induces an error of 33.47%. We also experiment MTZ to fully merge the hidden layers in the two LeNet-300-100 networks without any retraining *i.e.*, without line 10 in Algorithm 1. The averaged test error increases by only 1.50%.

Table 1 summarizes the errors of each LeNet pair before zipping ($\text{err}_A$ and $\text{err}_B$), after fully merged with retraining (re-$\text{err}_C$) and the number of retraining iterations involved (# re-iter). MTZ consistently achieves lossless network zipping on fully connected and convolutional networks, either they are dense or sparse, with 100% parameters of hidden layers shared. Meanwhile, the number of retraining iterations is approximately $19.0\times$ and $18.7\times$ fewer than that of training a dense LeNet-300-100 network and a dense LeNet-5 network, respectively.

Table 2: Test errors and retraining iterations of sharing all neurons (output layer fc8 excluded) in two well-trained VGG-16 networks for ImageNet and CelebA.

| Layer | $N_l^A$ | ImageNet (Top-5 Error) | | CelebA (Error) | | # re-iter |
|---|---|---|---|---|---|---|
| | | w/o-re-err$_C$ | re-err$_C$ | w/o-re-err$_C$ | re-err$_C$ | |
| conv1_1 | 64 | 10.59% | 10.61% | 8.45% | 8.43% | 50 |
| conv1_2 | 64 | 11.19% | 10.78% | 8.82% | 8.77% | 100 |
| conv2_1 | 128 | 10.99% | 10.68% | 8.91% | 8.82% | 100 |
| conv2_2 | 128 | 11.31% | 11.03% | 9.23% | 9.07% | 100 |
| conv3_1 | 256 | 11.65% | 11.46% | 9.16% | 9.04% | 100 |
| conv3_2 | 256 | 11.92% | 11.83% | 9.17% | 9.05% | 100 |
| conv3_3 | 256 | 12.54% | 12.41% | 9.46% | 9.34% | 100 |
| conv4_1 | 512 | 13.40% | 12.28% | 10.18% | 9.69% | 400 |
| conv4_2 | 512 | 13.02% | 12.62% | 10.65% | 10.25% | 400 |
| conv4_3 | 512 | 13.11% | 12.97% | 12.03% | 10.92% | 400 |
| conv5_1 | 512 | 13.46% | 13.09% | 12.62% | 11.68% | 400 |
| conv5_2 | 512 | 13.77% | 13.20% | 12.61% | 11.64% | 400 |
| conv5_3 | 512 | 36.07% | 13.35% | 13.10% | 12.01% | $1 \times 10^3$ |
| fc6 | 4096 | 15.08% | 15.17% | 12.31% | 11.71% | $2 \times 10^3$ |
| fc7 | 4096 | 15.73% | 14.07% | 11.98% | 11.09% | $1 \times 10^4$ |

Table 3: Test errors, number of shared neurons, and retraining iterations of adaptively zipping two well-trained VGG-16 networks for ImageNet and CelebA.

| Layer | $N_l^A$ | $\tilde{N}_l$ | ImageNet (Top-5 Error) | | CelebA (Error) | | # re-iter |
|---|---|---|---|---|---|---|---|
| | | | w/o-re-err$_C$ | re-err$_C$ | w/o-re-err$_C$ | re-err$_C$ | |
| conv1_1 | 64 | 64 | 10.28% | 10.37% | 8.39% | 8.33% | 50 |
| conv1_2 | 64 | 64 | 10.93% | 10.50% | 8.77% | 8.54% | 100 |
| conv2_1 | 128 | 96 | 10.74% | 10.57% | 8.62% | 8.46% | 100 |
| conv2_2 | 128 | 96 | 10.87% | 10.79% | 8.56% | 8.47% | 100 |
| conv3_1 | 256 | 192 | 10.83% | 10.76% | 8.62% | 8.48% | 100 |
| conv3_2 | 256 | 192 | 10.92% | 10.71% | 8.52% | 8.44% | 100 |
| conv3_3 | 256 | 192 | 10.86% | 10.71% | 8.83% | 8.63% | 100 |
| conv4_1 | 512 | 384 | 10.69% | 10.51% | 9.39% | 8.71% | 400 |
| conv4_2 | 512 | 320 | 10.43% | 10.46% | 9.06% | 8.80% | 400 |
| conv4_3 | 512 | 320 | 10.56% | 10.36% | 9.36% | 8.93% | 400 |
| conv5_1 | 512 | 436 | 10.42% | 10.51% | 9.54% | 9.15% | 400 |
| conv5_2 | 512 | 436 | 10.47% | 10.49% | 9.43% | 9.16% | 400 |
| conv5_3 | 512 | 436 | 10.49% | 10.24% | 9.61% | 9.07% | $1 \times 10^3$ |
| fc6 | 4096 | 1792 | 11.46% | 11.33% | 9.37% | 9.18% | $2 \times 10^3$ |
| fc7 | 4096 | 4096 | 11.45% | 10.75% | 9.15% | 8.95% | $1.5 \times 10^4$ |

## 4.2 Performance to Zip Two Networks (VGG-16) Pre-trained for Different Tasks

This experiment evaluates the performance of MTZ to automatically share information among two neural networks for different tasks. We investigate: *(i)* what the accuracy loss is when all hidden layers of two models for different tasks are fully shared (in purpose of maximal size reduction); *(ii)* how much neurons and parameters can be shared between the two models by MTZ with at most $0.5\%$ increase in test errors allowed (in purpose of minimal accuracy loss).

**Dataset and Settings.** We explore to merge two VGG-16 networks trained on the ImageNet ILSVRC-2012 dataset [24] for object classification and the CelabA dataset [16] for facial attribute classification. The ImageNet dataset contains images of $1,000$ object categories. The CelebA dataset consists of 200 thousand celebrity face images labelled with 40 attribute classes. VGG-16 is a deep convolutional network with 13 convolutional layers and 3 fully connected layers. We directly adopt the pre-trained weights from the original VGG-16 model [25] for the object classification task, which has a $10.31\%$ error in our evaluation. For the facial attribute classification task, we train a second VGG-16 model following a similar process as in [17]. We initialize the convolutional layers of a VGG-16 model using the pre-trained parameters from imdb-wiki [23], then train the remaining 3 fully connected layers till the model yields an error of $8.50\%$, which matches the accuracy of the VGG-16 model used in [17] on CelebA. We conduct two experiments with the two VGG-16 models. *(i)* All hidden layers in the two models are $100\%$ merged using MTZ. *(ii)* Each pair of layers in the two models are adaptively merged using MTZ allowing an increase ($< 0.5\%$) in test errors on the two datasets.

Table 4: Test errors of pre-trained single ResNets and the joint network merged by MTZ. $1\times$ is the number of parameters of one single ResNet excluding the last classification layer.

| | #par. | C100 | GTSR | OGlt | SVHN | UCF | mean |
|---|---|---|---|---|---|---|---|
| $5\times$ Single model | $5\times$ | 29.19% | 1.48% | 14.40% | 6.86% | 37.83% | **17.95%** |
| Joint model | $1.5\times$ | 29.13% | 0.09% | 15.65% | 7.08% | 39.04% | **18.20%** |

**Results.** Table 2 summarizes the performance when each pair of hidden layers are $100\%$ merged. The test errors of both tasks gradually increase during the zipping procedure from layer `conv1_1` to `conv5_2` and then the error on ImageNet surges when `conv5_3` are $100\%$ shared. After $1,000$ iterations of retraining, the accuracies of both tasks are resumed. When $100\%$ parameters of all hidden layers are shared between the two models, the joint model yields test errors of $14.07\%$ on ImageNet and $11.09\%$ on CelebA, *i.e.*, increases of $3.76\%$ and $2.59\%$ in the original test errors.

Table 3 shows the performance when each pair of hidden layers are adaptively merged. Ultimately, MTZ achieves an increase in test errors of $0.44\%$ on ImageNet and $0.45\%$ on CelebA. Approximately $39.61\%$ of the parameters in the two models are shared ($56.94\%$ in the $13$ convolutional layers and $38.17\%$ in the $2$ fully connected layers). The zipping procedure involves $20,650$ iterations of retraining. For comparison, at least $3.7 \times 10^5$ iterations are needed to train a VGG-16 network from scratch [25]. That is, MTZ is able to inherit information from the pre-trained models and construct a combined model with an increase in test errors of less than $0.5\%$. And the process requires at least $17.9\times$ fewer (re)training iterations than training a joint network from scratch.

For comparison, we also trained a fully shared multi-task VGG-16 with two split classification layers jointly on both tasks. The test errors are $14.88\%$ on ImageNet and $13.29\%$ on CelebA. This model has exactly the same topology and amount of parameters as our model constructed by MTZ, but performs slightly worse on both tasks.

### 4.3 Performance to Zip Multiple Networks (ResNets) Pre-trained for Different Tasks

This experiment shows the performance of MTZ to merge more than two neural networks for different tasks, where the model for each task is pre-trained using deeper architectures such as ResNets.

**Dataset and Settings.** We adopt the experiment settings similar to [21], a recent work on multi-task learning with ResNets. Specifically, five ResNet28 networks [30] are trained for diverse recognition tasks, including CIFAR100 (C100) [12], German Traffic Sign Recognition (GTSR) Benchmark [27], Omniglot (OGlt) [13], Street View House Numbers (SVHN) [20] and UCF101 (UCF) [26]. We set the same 90% compression ratio for the five models and evaluate the performance of MTZ by the accuracy of the joint model on each task.

**Results.** Table 4 shows the accuracy of each individual pre-trained model and the joint model on the five tasks. The average accuracy decrease is a negligible $0.25\%$. Although ResNets are much deeper and have more complex topology compared to VGG-16, MTZ is still able to effectively reduce the overall number of parameters, while retaining the accuracy on each task.

## 5 Conclusion

We propose MTZ, a framework to automatically merge multiple correlated, well-trained deep neural networks for cross-model compression via neuron sharing. It selectively shares neurons and optimally updates their incoming weights on a layer basis to minimize the errors induced to each individual task. Only light retraining is necessary to resume the accuracy of the joint model on each task. Evaluations show that MTZ can fully merge two VGG-16 networks with an error increase of $3.76\%$ and $2.59\%$ on ImageNet for object classification and on CelebA for facial attribute classification, or share $39.61\%$ parameters between the two models with $< 0.5\%$ error increase. The number of iterations to retrain the combined model is $17.9\times$ lower than that of training a single VGG-16 network. Meanwhile, MTZ is able to share 90% of the parameters among five ResNets on five different visual recognition tasks while inducing negligible loss on accuracy. Preliminary experiments also show that MTZ is applicable to sparse networks. We plan to further investigate the integration of MTZ with weight pruning in the future.

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
