[Reviews · NeurIPS 2018]

Reviewer 1



The paper considers the following problem: given a collection of pretrained deep models for different tasks, how to obtain a shared/joint compressed model? Authors work under the assumption that all models have same architecture and propose an approach to identify hidden units across corresponding layers in these models that can be merged together. The criterion for merging looks at the errors introduced in the pre-activations of the units when their corresponding incoming weights are merged -- hidden unit pairs (for the case of two pretrained networks) which result in lowest error are merged. Experiments with LeNets on MNIST and VGG-16 on Imagenet and CelebA are reported to show the effectiveness of the method. Strengths: The problem setting considered in the paper seems novel as well as reasonable from a practical perspective. Weaknesses: The empirical evaluation of the proposed method is not comprehensive enough in my view - (i) some natural baselines are not considered, eg, the paper starts with trained theta_1... theta_n and merges them in a shared model (theta) - a natural alternative is start with a shared model (either fully shared or usual branching architectures) and train it jointly on all tasks -- how does it perform in comparison to the proposed method on the error vs model-size plane? (ii) Experiments in Sec 4.1 (two LeNet models, both trained on MNIST) do not convey much information in my view. I am missing the point in merging two models trained on the same task (even with different seeds). (iii) Experiments in sec 4.2 are done the VGG 16 which is a quite heavy model to begin with. It's hard to judge the significance of the results due to lack of baselines. Authors should show model size and accuracy numbers for some recently proposed multitask deep models (for the same set of tasks - imagenet and celebA) to make it easier for the reader. (iv) All experiments and even the formulation in 3.3 is for two models. How does the method extend to more than two? (v) Algorithm 1 refers to d(w^A, w^B) and f(w^A, w^B). Where are they defined in Sec 3.3? I couldn't find a definition for these. (vi) The paper claims that retraining needed after the merging is quite fast. What is the time needed for merging? From Algorithm 1, it looks like the method needs L forward passes for all training samples to merge L layers (as after merging a layer 'i', the activations for layer 'i+1' need to be recomputed)? =================================== I looked at the author response and I still feel many of my concerns are not addressed satisfactorily, particularly the following points that I raised in my initial review. (i) Natural baseline: I am not fully convinced about the reasons the authors cite for not including this baseline in the first draft. If the results of the baseline were worse than MTZ, it would've been still useful to include the results in the paper and discuss/explain why it performs worse than MTZ in the case of fully shared model. I think this is an interesting observation on its own which deserves discussion, since one would expect joint training of fully shared model to perform better than the two step optimization process of first training individual models and then merging them. (ii) Same task with different random seeds: I am willing to buy the rationale given in the author response for this. However I would've still liked to see experiments for this setting using more recent architectures that authors use for experiments in Sec 4.2 with same datasets (ideally, ResNets or something more recent..). It is not clear why the paper uses LeNets + mnist. (iii) More recent architectures and other competitive baselines: Author response has some results on MTZ applied on ResNets. Again, the fully shared baseline is missing from the table. My other point about comparing with recent multitask approaches (which also result in a compressed model) still remains unaddressed. It is difficult to assess the effectiveness of the proposed method (in terms of the trade-off it provides b/w final model size and accuracy) in the absence of any comparison. The absence of any baselines is perhaps the weakest point of the current draft in my view. (iv) (v) I am fine with the explanation provided in the author response for these. (vi) Number of fwd passes: If I understand correctly, to apply the approach with a single forward pass, one needs to store *all* the activations of previous layer in the memory so that these can be fwd-propagated after zipping the current layer. I couldn't find details in the paper on how many input examples are used for the zipping process. Author response just mentions "Moreover, only a small part of the whole training dataset is needed to generate an accurate enough Hessian matrix as mentioned in [8].", without giving details on number of examples used in the experiments. Algorithm 1 "input" on the other hand mentions "training datum of task A and B" which gives a feeling that full data is used during MTZ.

Reviewer 2



This paper tackles the relevant problem of cross-model compression for applications with memory limitations. To do so, a multi-task zipping (MTZ) approach is introduced, which automatically and adaptively merges deep neural networks via neuron sharing. The method iteratively decides on a layer basis and based on a computed functional difference, which pairs of neurons to merge and how to update their weights in order to mitigate any potential error increase resulting from the compression. A proof of concept is performed on MNIST for fully connected, convolutional and sparse networks; and additional results are reported on image classification tasks (ImageNet vs CelebA pre-trained models). The presentation of the paper is clear and ideas are easy to follow. In the related work section, authors discuss knowledge distillation based approaches, referencing [2][3][15]. Although [2][3] are indeed model compression methods, they are not rooted in the knowledge distillation approach introduced in [15]. However, FitNets (https://arxiv.org/pdf/1412.6550.pdf), which does extend the knowledge distillation idea of [15] in the context of model compression is not referenced nor discussed. Please cite all of those works appropriately. The residual adapters modules and their motivation (https://arxiv.org/pdf/1705.08045.pdf) seem fairly related to the MTZ approach; the goal being to build a single network to learn multiple visual domains while keeping the number of domain specific parameters low. However, authors fail to discuss this paper. It would be beneficial to compare the MTZ results to the ones reported in this paper, and for some overlapping tasks. In Equation (5), what is the influence of alpha? How do you set it in the experimental section? What is the impact of the value of this hyper-parameter? In Algorithm 1, a light retraining is computed after merging each layer. What is the effect of re-training after each merging? What would be the impact in performance of re-training only once after the whole merging procedure is completed? In tables reporting results, does the number of re-training iterations include all the intermediate retraining steps? It might be worth adding to the results (1) the compression rate of the merged model w.r.t to the smaller single model in terms of number of parameters (or operations), (2) the compression rate w.r.t. all the models prior to zipping, (3) number of domain specific vs number of domain agnostic parameters in the final networks, (4) report the number of re-training iterations w.r.t. re-training the full model (speed-up instead of number of iterations itself). There is no mention about intentions on making the code publicly available. So far, the model has been tested to merge networks with the same number of layers and which are relatively shallow. However, this seems to have rather limited applicability (networks trained for different tasks may eventually have different number of layers). Have the authors considered extending the framework to networks of different depth? Moreover, it might be worth testing the approach on a more recent (deeper) state-of-the-art network, such as resnet. It would be more informative to report the plots on Figure 2 for the ImageNet vs CelebA experiments, since for MNIST it is hard to perceive any differences while augmenting the number of shared neurons (in the MTZ case). Have the authors tried to zip more than 2 networks? That would be an interesting experiment to compare to the residual adapters on several visual tasks. In the experimental section 4.2, how was the number of shared neurons per layer selected? Finally, in lines 262-264, it seems that the number of parameters shared for fully connected layers is lower than in convolutional layers. However, it is worth mentioning that fully connected layers are usually the ones containing most of the parameters. ----------------------------- I thank the authors for their detailed answer; addressing many of concerns raised by the reviewers. The paper could still benefit from a proper comparison with the multi-task baseline.

Reviewer 3



This paper proposes Multi-Task Zipping (MTZ), a framework to automatically and adaptively merge correlated, well-trained deep neural networks for cross-model compression via neuron sharing. It decides the optimal sharable pairs of neurons on a layer basis and adjusts their incoming weights such that minimal errors are introduced in each task. Evaluations show that MTZ is able to merge parameters across different architectures without increasing the error rate too much. I liked this submission and the motivation of neuron sharing for the compression of multiple trained deep neural networks where the paper focuses on the inter-redundancy of multiple models. The layer-wise computation for the Hessian-based difference metric nicely solves the computational overload. Details: (1) Can this approach be used in other architectures such as RNNs in sequence modeling? The experiments on LeNet is relatively toy. (2) How to extend this approach to multiple models (>=3)? What if we have two models with different sizes and layers?